# Evolution Regularity of Continuous Surface Structures Shaped by Laser-Supported Fictive-Temperature Modifying

**Wei Liao, Chuanchao Zhang, Jing Chen, Ke Yang, Lijuan Zhang, Xiaolong Jiang, Yang Bai, Haijun Wang, Xiaoyu Luan, Xiaodong Jiang, Xiaodong Yuan, Wanguo Zheng and Qihua Zhu \***

Laser Fusion Research Center, China Academy of Engineering Physics, Mianyang 621900, China; zhchch@caep.cn (C.Z.)
\* Correspondence: qihzh@163.com

**Abstract:** The influence of residual heat on the fictive temperature modification zone of fused silica for different $CO_2$ laser scanning time intervals was investigated to precisely control the profiles of hydrofluoric (HF) acid-etched fused silica surface, which were formed by the increasing HF acid-etching rate for fused silica with increasing fictive temperature induced by $CO_2$ laser scanning. The surface profiles of HF acid-etched fused silica treated by different scanning time intervals of $CO_2$ laser were measured by a stylus profilometry, and experimental results indicate that the $CO_2$ laser scanning time intervals intensively influence the HF acid-etched surface profiles of fused silica. The increasing depth of surface profiles treated by shorter scanning time intervals shows that the fictive temperature modification zone significantly expands. Numerical simulations of the fictive temperature modification zone induced by different scanning time intervals indicate that the residual heat of $CO_2$ laser scanning with shorter time intervals leads to a dramatical increase in the fictive temperature modification zone. By adjusting the residual heat of $CO_2$ laser scanning intervals, various surface profiles of fused silica can be obtained by HF acid-etching of fused silica.

**Keywords:** $CO_2$ laser; overlapped scanning; residual heat; continuous surface profile





## 1. Introduction

Continuous phase plates (CPP) with continuous surface profiles play an important role in shaping of focal spot for inertial confinement fusion (ICF)-driven laser devices [1,2]. The National Ignition Facility (NIF) [3,4] in USA and the Laser MégaJoule (LMJ) in France both use CPP to obtain a pre-defined shape and a homogeneous energy distribution of focal spot for improving symmetry of center-ward compression of target. The CPP, as a phase-modulated optical shaping element, achieves far-field beam shaping by changing the phase distribution in the near-field. The special surface profile of CPP is non-periodic, smooth, continuously undulating, and with large gradients, which makes the fabrication of CPP by conventional optical processing technologies very challenging. Therefore, the research of CPP fabricating technology is a hot topic in the field of optical manufacturing [5–8].

Currently, most of the CPPs used in ICF-driven laser facilities are fabricated by magnetorheological finishing (MRF), which was first proposed in 2003 by Menapace et al. [9,10]. The MRF technique is a deterministic processing technique, which can be used to shape an arbitrary surface profile by the convolution of the removal function and the dwell time, and ideally suitable for the fabricating of CPPs [11,12]. However, due to the limit of the size of MRF tool head, it is difficult to achieve the smallest CPP unit below 3 mm [13]. Moreover, MRF works in the plastic deformation zone of the material, which makes the machining accuracy very high; however, the MRF is time-consuming and expensive. In past decades, a number of deterministic processing methods have been explored for CPP fabrication. For example, Xu et al. [14,15] proposed the use of ion beam figuring (IBF) technology to fabricate CPP, which has the advantage to achieve a smaller spatial period of 0.7 mm and a

high fabrication accuracy of 10 nm RMS at a surface gradient of 1.8 μm/cm. Su et al. [16] proposed the use of atmospheric pressure plasma processing (APPP) for the fabrication of CPP, which has a high fabrication efficiency of 9.3 h for a 320 mm diameter piece of CPP with a good machining accuracy.

However, with the increase in output energy of ICF-driven laser facilities [17,18], there are two important issues for CPP fabricated by the reported fabrication technologies, such as MRF, IBF, and APPP. The first is the laser-induced damage of CPP under extremely high-power laser irradiation, which is caused by contaminants during the fabrication process [19–21]. At present, the problem of damage to the fused silica elements, typically represented by CPP, is the bottleneck limiting the output energy increase in the ICF-driven devices. The other is the minimization of fabrication cost for CPP. The reported fabrication technologies can not satisfy the above demands of CPP. Therefore, an inexpensive method of fabricating CPP with high damage resistance is still needed.

The surface profile fabrication method based on the difference of HF acid-etching rates for fused silica with different fictive temperatures is promising to economically and effectively fabricate CPP. This method is based on the principle that the fused silica material will experience sudden heating and cooling due to the short irradiation of the laser. The volume contraction of the fused silica during rapid cooling reduces the bond energy of the Si-O-Si bond, which makes it easier for a chemical reaction to take place [22]. For the proposed method, the local fictive temperature distribution of fused silica is modified by $CO_2$ laser irradiation [23], and then the $CO_2$ laser-treated fused silica samples are etched by HF acid and the wanted surface profile is formed [24,25]. This method has the advantages of low cost, widely adjustable processing dimensions, and high damage resistance, especially when the etching step is performed using the advanced mitigation processing (AMP) [26,27]. Zhang et al. [28] has reported the fabrication of concave micro-lens arrays by this method. However, little work has been reported on the use of this method for fabricating non-periodic and continuously undulating surface profiles. Therefore, the proposed idea of fabricating low-cost, high damage threshold CPPs by a combination of laser irradiation modification and chemical etching has never been mentioned in previous studies and is highly innovative. However, one of the biggest challenges affecting the practical application of the method was found during the conduct of the research, in which the residual heat accumulated between the $CO_2$ laser irradiations of fused silica samples, which makes the local fictive modification zone uncontrollable.

Therefore, in this study, we investigate the effect of residual heat between the scans on the local fictive temperature distributions. The evolution of HF acid-etched surface profiles with different scan time intervals and different scan overlap rates was investigated by experiments and simulations, respectively. The residual heat from the previous laser scan will transfer into the body of the material, and the thermodynamic temperature in the irradiated zone gradually decreases. As the residual heat diffuses, the amount of change in the modulated zone produced by the subsequent scan becomes smaller. When the scan time interval exceeds 30 s, the profiles shaped by the two scans can remain the same in the case of partially overlapping scans, which makes the removal function more consistent and the final profile more deterministic.

## 2. Materials and Methods

All samples used in this study were planar Corning 7980 fused silica components with dimensions of 50 × 50 × 4 mm. The samples were cleaned with deionized water and ethanol under ultrasonic agitation, respectively.

The experimental setup used for the $CO_2$ laser scanning of fused silica samples is shown in Figure 1. The $CO_2$ laser operates in continuous mode with a maximum output power of 100 watts and a power stability of better than ±3%. The power of output laser is first regulated by a commercial acousto-optic modulator, and the regulated power is monitored in real time by a power meter. The laser beam is expanded by a beam expander and sent into the galvanometer scanner. Then, the expanded beam is focused by an F-Theta

lens with a focal length of 100 mm, and the diameter of the focal spot on the sample surface is about 90 μm. During laser scanning, the sample is placed on a two-dimensional translation table for changing the scanning area. The laser scan paths in this study were all set to be 10 mm straight lines. The laser power treating the sample was set to be 50 W. The scanning velocity is set to 1.4 m/s.

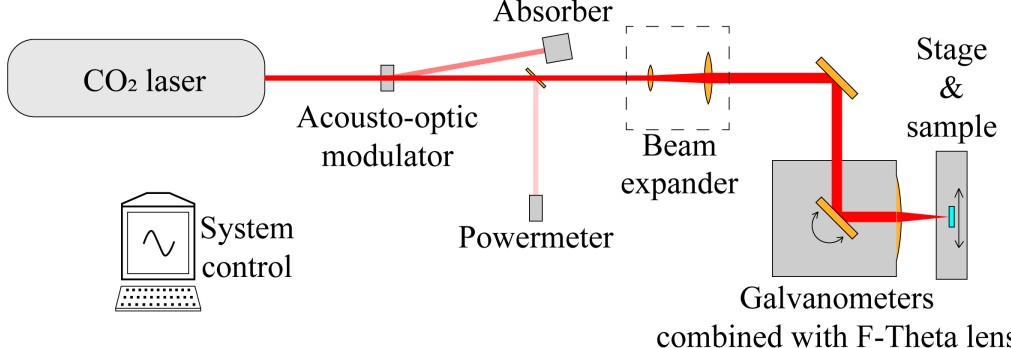

**Figure 1.** Experimental setup for laser scanning.

To investigate the influence of residual heat on the same scanning line on fused silica sample, repeated $CO_2$ laser scanning experiments with different time intervals on the same position of fused silica samples were divided into two groups. One group was with scanning time interval of 0.1 s. The other group was with scanning time interval of 30 s. For each group, ten lines were scanned on fused silica samples with increasing repeat scans from one to ten.

To investigate the influence of residual heat on the adjacent scanning lines, 10% and 30% overlaps of $CO_2$ laser scanning lines on fused silica samples were performed with scanning time intervals of 0.1 and 30 s, respectively.

After the $CO_2$ laser treatment, the fused silica samples were etched in an HF acid buffer with a weight fraction of 2.0% HF, 12.0% $NH_4F$, and 86.0% deionized water for 30 min under ultrasonic stirring conditions at 1.3 MHz. Thereafter, the samples were rinsed using deionized water and allowed to air dry. The HF acid-etched surface profiles were measured using a stylus profilometry.

## 3. Results and Discussion

### 3.1. HF Acid-Etched Surface Profiles of Repeated Scanning Lines

Figure 2a,b shows the HF-acid etched section profiles in the middle of repeated scanning lines on fused silica samples with increasing repeat scans from one to ten for the scanning time intervals of 0.1 and 30 s, respectively. It can be seen that the evolution trend of the HF-acid etched surface profiles with increasing number of scans intensively depends on the scanning time interval. For the scanning time interval of 0.1 s, the size of HF-acid etched profiles of repeat scan lines on fused silica increases significantly with the increasing number of scans. In contrast, for the scanning time interval of 30 s, the HF-acid etched profiles of repeat scan lines on fused silica increases a little with increasing number of scans. The regularity of evolution trend can be shown more clearly from the curves of depth and width of HF-acid etched repeat scan lines on fused silica with increasing number of scans, as shown in Figure 2c,d. For the scanning time interval of 0.1 s, the curves of depth and width of HF-acid etched profiles with increasing number of scans is similar to a conic curve, which increase rapidly for the first four scans, and then gradually slow down after the 5th scan. The depth of HF-acid etched profiles increases from 0.87 μm in one scan to 2.68 μm in ten scans. The width of HF-acid etched profiles increases from 35 μm in one scan to 75 μm in ten scans. The depth and width of HF-acid etched profiles increase approximately 3.1 and 2.1 times from one scan to ten scans, respectively. In addition, the HF-acid etched profiles continue increasing in size despite the number of scans reaching ten. In contrast,

for the extended scanning time interval of 30 s, the depth and width of the HF-acid etched profiles remain almost constant after more than four scans. The depth of HF-acid etched profiles increases from 0.91 µm in one scan to 1.30 µm in ten scans. The width of HF-acid etched profiles increases from 37 µm in one scan to 42 µm in ten scans. The depth and width of HF-acid etched profiles increase approximately 1.4 and 1.1 times from one scan to ten scans, respectively.

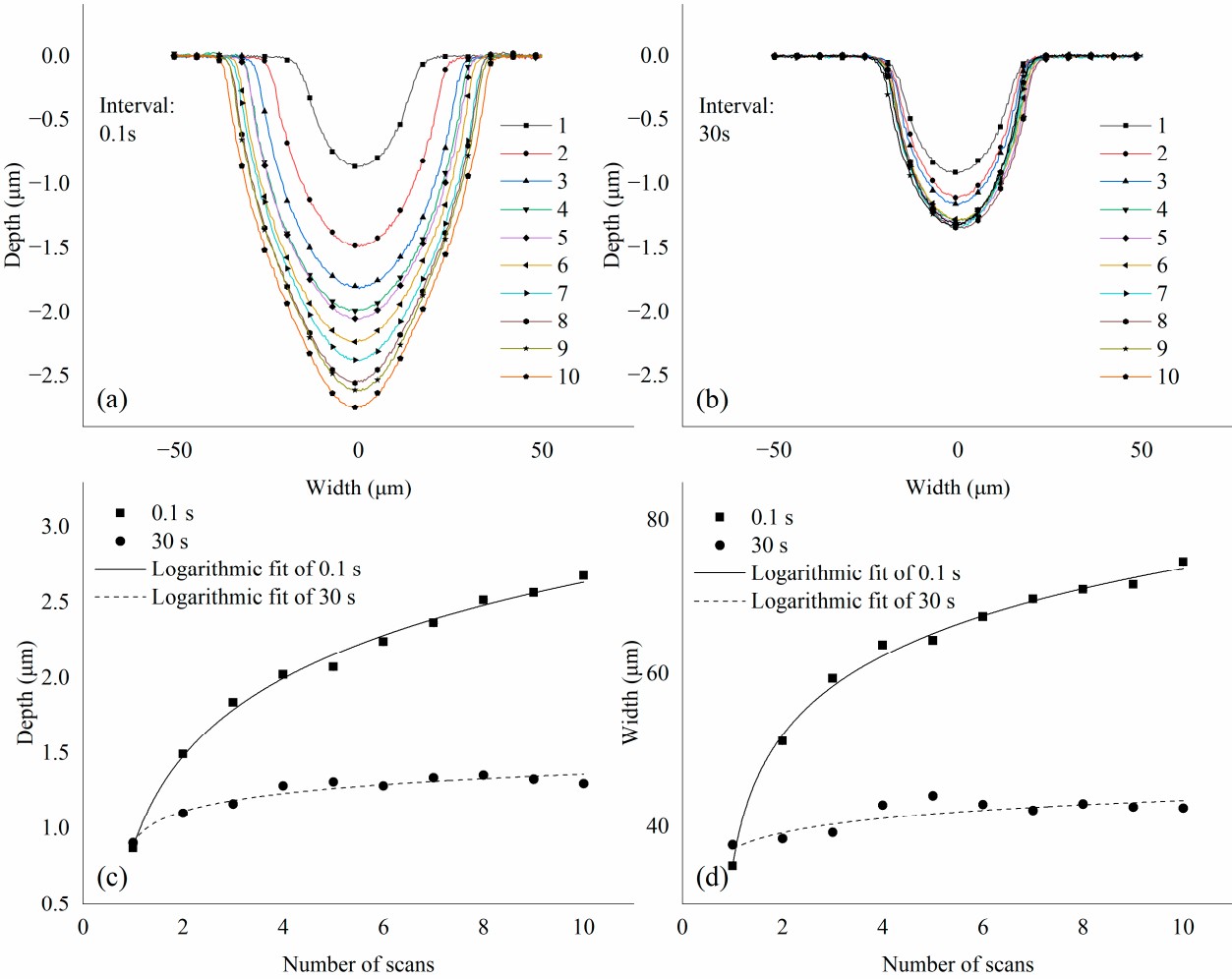

**Figure 2.** The evolution of HF-acid etched section profiles on fused silica with increasing repeat scans for the scanning time intervals of 0.1 s (**a**) and 30 s (**b**), respectively. HF-acid etched depth (**c**) and width (**d**) with the increasing number of scans on fused silica.

### 3.2. HF Acid-Etched Surface Profiles of Adjacent Scanning Lines

Figure 3a exhibits the HF-acid etched section profiles of adjacent scanning lines on fused silica for the scanning time intervals of 0.1 s. It can be observed that, although the scanning parameters of $CO_2$ laser are exactly the same for the adjacent two lines, the second scan results in a significantly larger HF-acid etched profile than the first scan, again demonstrating that the residual heat from the first scan at short time intervals can significantly affect the fictive temperature distribution of the second scan. On the other hand, when the spatial overlap is 10%, the groove depth is approximately 0.755 µm at the first scan position and 1.34 µm at the second scan position. When the spatial overlap rate is increased to 30%, the groove depth is approximately 0.749 µm at the first scan position and 1.36 µm at the second scan position. These results show that, in contrast to the time interval parameter, the spatial overlap rate parameter does not have a significant effect on the contour dimensions.

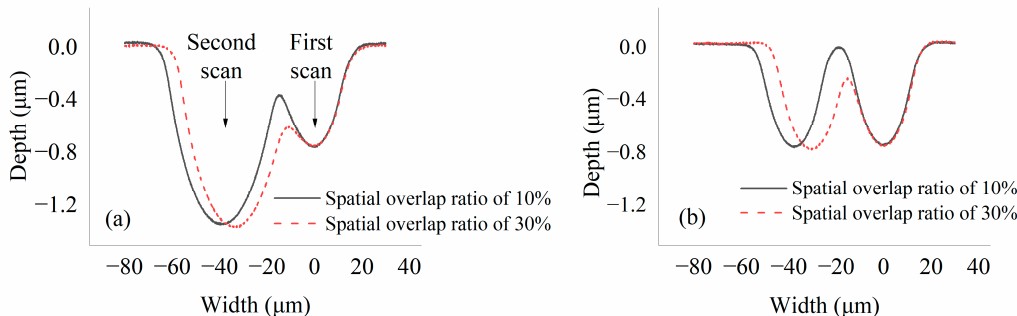

**Figure 3.** Structural profiles scanned at different spatial overlap rates for time intervals of (**a**) 0.1 s and (**b**) 30 s, respectively. In the graph, 10% and 30% indicate the spatial overlap ratios between the two scan lines, respectively.

In contrast, for the scanning time interval of 30 s, the HF-acid etched surface profiles of the adjacent two lines remain constant, regardless of whether the spatial overlap ratios are 10% or 30%, as shown in Figure 3b. A possible mechanism for this regulation is that when the scanning time interval becomes longer, most of the residual heat after the previous laser irradiation can diffuse into the substrate or the ambient conditions through heat conduction. In this way, the thermodynamic temperature of the local material can be restored to its initial state at the time of subsequent laser irradiation.

Comparing the regularities of surface profile evolution trends for HF-acid etched repeat scan lines on fused silica with increasing number of scans, it is clear to see that the evolution of the fictive temperature distribution is very complex for repeated irradiation of fused silica using a $CO_2$ laser. First, repeated scanning at very short time intervals leads to a significant increase in the size of the formed profile. This phenomenon is somewhat similar to conventional deterministic processing methods. However, the difference is that, in addition to the increase in depth, the lateral dimensions increase significantly and there is a non-linear relationship between these two changes with the number of irradiations. Second, when the scanning time interval is extended, the dimension of the formed profile becomes affected very slightly by the number of scans. Clearly, the above phenomenon will lead to a poor predictability when using the proposed method to fabricate CPP-like surface profiles. Therefore, we believe that it is more appropriate to complete the entire surface fictive temperature regulation in a single pass when actually fabricating the profile. Specifically, the laser scanning path should be designed in a way that the desired fictive temperature distribution can be achieved in a single scan, and if overlapping scans are necessary, the time interval between two passes should be maximized.

### 3.3. Simulation of Fictive Temperature Distribution for Overlapped Scanning

To better understand the influence of residual heat on the fabricated profile, the fictive temperature distribution of fused silica under overlapping laser scans with different parameters has been numerically simulated. The finite element method was used for the simulations and the model was constructed as two-dimensional to simplify the computations, as shown in Figure 4. Considering that the model only involves physical fields of heat-solid coupling, commercial finite element analysis software was used for the simulation. To contain sufficient heat in order that the boundary conditions can be simplified, the dimensions of the fused silica material are set to 2 × 1 mm. The laser is incident on the upper surface; therefore, the boundary conditions on the upper surface are set to radiation heat emission and the boundary conditions on the left, right, and lower surfaces are set to heat fluxes. The grid is divided into non-uniform meshes and finer meshes for the laser irradiated area, taking into account the large temperature gradients in the laser irradiated area.

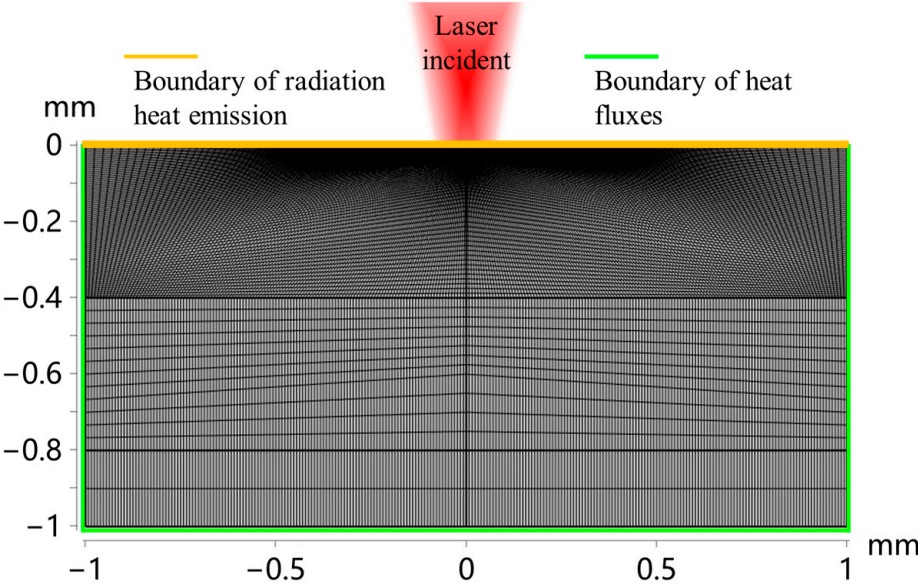

**Figure 4.** Two-dimensional numerical model for simulation calculations.

Provided that the laser irradiation does not lead to drastic evaporation of the material, the thermodynamic temperature of the irradiated fused silica follows the classical heat transfer equation, as follows:

$$\rho c \frac{\partial T(x,y,z,t)}{\partial t} = k\nabla^2 T(x,y,z,t) + Q(x,y,z,t)$$

where, $\rho$, $c$, and $k$ are the density, heat capacity, and thermal conductivity of fused silica, respectively. $T$ is the temperature field distribution of the material and $Q$ is the heat source introduced by the laser irradiation.

We have chosen a laser power that leads to a local thermodynamic temperature slightly below 3000 K, at which the evaporation of the fused silica material is not significant and the three key coefficients of the heat transfer equation—density, heat capacity, and thermal conductivity—are accurately given. Here, the laser irradiation is simplified in the form of a bulk heat source with an exponential decay along the axial direction, where the exponent is the absorption coefficient of the $CO_2$ laser by the fused silica material. As the laser is a Gaussian spot, the bulk heat source is set to a Gaussian distribution along the radial direction. Considering that the laser has a scanning speed of 1.4 m/s and a spot diameter of 90 μm, the duration of laser irradiation experienced at a location on the sample surface is set to 64 μs.

Based on the above numerical model, the local thermodynamic temperature distribution can first be calculated for the laser irradiation and the subsequent natural cooling process. The local thermodynamic temperature distribution at the moment of 0.001 s after laser irradiation is given in Figure 5. Moreover, the graph gives the thermodynamic temperature profile as a function of time at the center of laser irradiation, which can be considered as the highest temperature point of the material during laser irradiation. The curve shows that the thermodynamic temperature of the material experiences a sudden increase and a rapid decrease. The highest temperature during the entire process reaches over 2800 K, which is above the softening point temperature of the fused silica material of approximately 1300 K. After experiencing these high temperatures, the structure of fused silica changes as follows. When laser irradiation is present, the thermodynamic temperature of the fused silica gradually increases until it is above the softening point, at which point the viscosity of the material decreases and plastic deformation occurs. When the laser irradiation stops, the thermodynamic temperature of the fused silica drops rapidly and the material, which has been plastically deformed, creeps in the opposite direction again. However, during

the cooling process, the fused silica becomes more viscous and less fluid, and the rate of relaxation of the local structure cannot catch up with the change rate of the thermodynamic temperature. Therefore, when the thermodynamic temperature of the fused silica finally drops below the softening point, the local structure is "frozen" at some higher temperature value, which is defined as the fictive temperature of the fused silica.

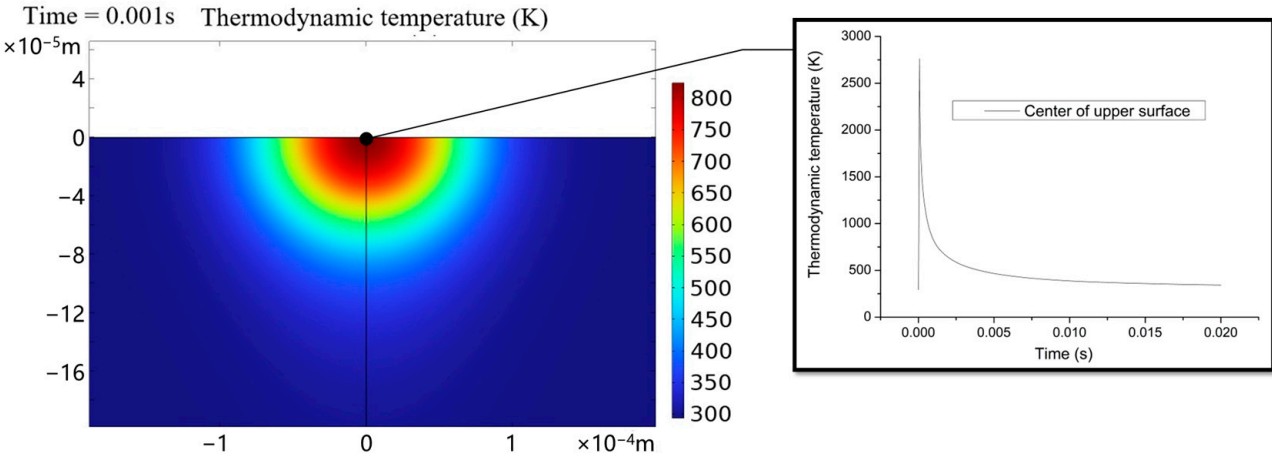

**Figure 5.** Thermodynamic temperature distributions of 0.001 s after laser irradiation, and thermodynamic temperature change curve at the center of the laser irradiation zone.

Considering that the rate of structural relaxation is proportional to the difference between the fictive and thermodynamic temperatures and inversely proportional to the relaxation time constant of the material at the current thermodynamic temperature [29], the distribution of the local fictive temperature field can be calculated simultaneously with the thermodynamic temperature. Therefore, we added a second laser exposure at the same location at the end moments of the different cool-down times—0.01, 0.1, 1, 10, and 100 s, respectively. The specific effect of residual heat can be seen in the difference between the ranges of the fictive temperature modulation zones. Figure 6a gives the distribution of the local fictive temperature after the first laser irradiation and Figure 6b–f gives the distribution of the local fictive temperature after the second laser irradiation at different time intervals. Figure 6g gives a fictive temperature distribution curve for the center of the incident point along the depth direction. It can be seen that after the first laser irradiation, the maximum depth of the fictive temperature regulation zone is approximately 7 µm. When a second irradiation is performed at a short time interval after the first laser irradiation, e.g., 0.01 s, the maximum depth of the fictive temperature regulation zone reaches approximately 9 µm, an increase of approximately 2 µm compared to the first time. When gradually increasing the time interval between irradiations, a significantly smaller increase in the maximum depth of the fictive temperature regulation zone can be seen. This would suggest that the longer the time interval, the smaller the variation in the range of the fictive temperature modulation zone after the second laser irradiation. Since the next forming process is based on the difference in reaction rate between the fused silica material and HF acid at different imaginary temperatures, this indicates that the longer the time interval, the smaller the expansion of the profile formed by the etching. Expectedly, the trends given by the numerical simulations are consistent with the experimental results presented in the previous section.

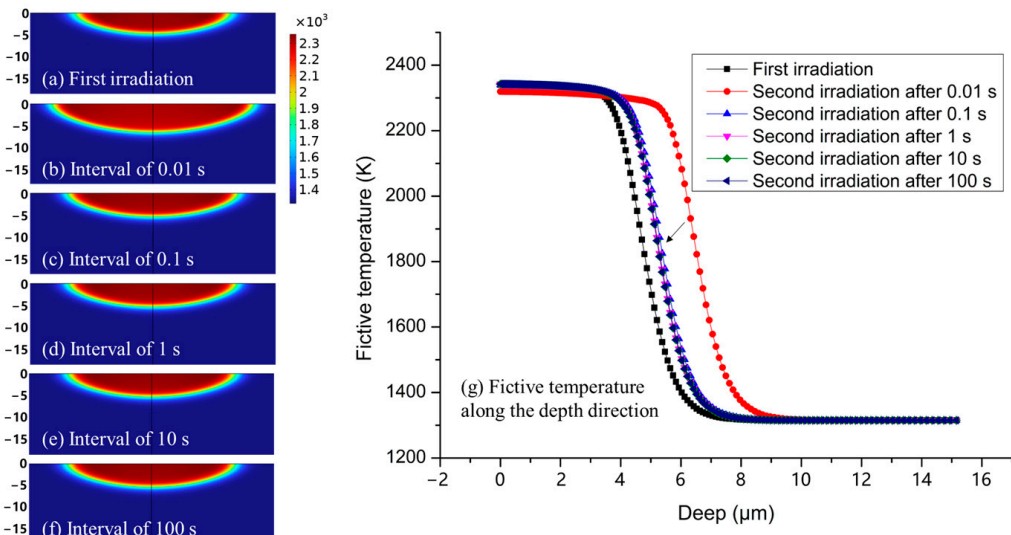

**Figure 6.** Fictive temperature distribution after (**a**) the first laser irradiation and after the second laser irradiation at time intervals of (**b**) 0.01 s, (**c**) 0.1 s, (**d**) 1 s, (**e**) 10 s, and (**f**) 100 s, respectively. (**g**) The fictive temperature distribution curve for the center of the incident point along the depth direction.

Based on the previous understanding, we can then further simulate the residual heat effect during partial overlap scanning. The spatial overlap between the two laser scans is set to 30%, with time intervals of 0.01 and 100 s. The fictive temperature distributions resulting from the two scans at different time intervals are given in Figure 7a,b. It is very clear from the figure that the short interval time leads to a significant increase in the fictive temperature distribution for the second scan, while the fictive temperature distribution for the two scans remains essentially the same for the long interval time. It is clear that residual heat is the key reason for these differences. To see the effect of residual heat more directly, the following considerations were made: Since all four boundaries of the model are non-thermally insulated, the injected laser energy will dissipate from all four boundaries over time, in order that the integral value of the model's full domain temperature field can represent the residual heat remaining in the domain. Figure 7c gives the variation of the normalized temperature integral over the modeling domain for different intervals. From the figure, it can be seen more visually that the temperature integral increases significantly after laser irradiation, indicating that a large amount of heat has accumulated locally. If a second irradiation is carried out within a short period of time, residual heat will accumulate in the body, whereas extending the time interval will allow sufficient time for the material to absorb the residual heat between laser irradiations, and thus maintain the consistency of the fictive temperature modulation zone. In this case, there is a linear correlation between the final profile and the laser irradiation parameters, which makes continuous surface treatment of large areas feasible.

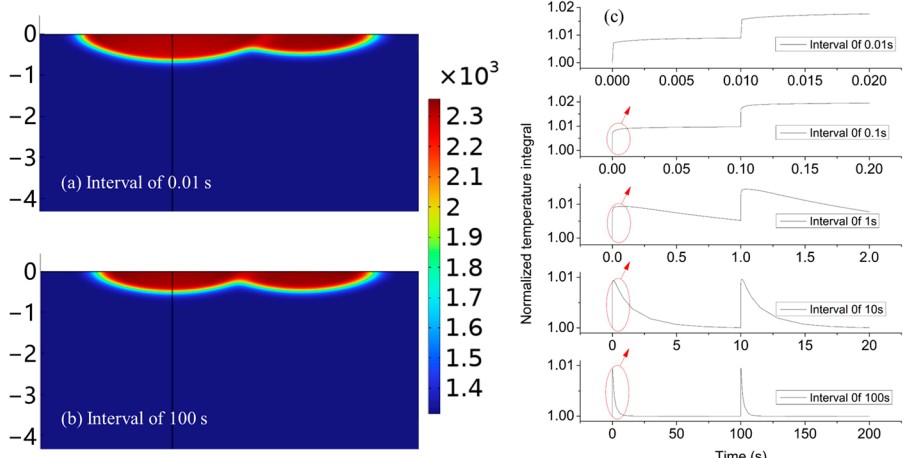

**Figure 7.** Fictive temperature distribution after two laser irradiations with time intervals of (**a**) 0.001 s and (**b**) 1 s at a spatial overlap of 30%. (**c**) Normalized temperature integral over the modeling domain for different time intervals, in which the duration of the above curve is the same as the duration marked with a red circle on below curve.

## 4. Conclusions

The new method of using $CO_2$ laser irradiation to modulate a fictive temperature distribution followed by HF acid etching to create a continuous surface profile on a fused silica surface has significant potential for the fabrication of CPP. However, the residual heat of the material under multiple laser scans can lead to an inconsistent range of fictive temperature modulation zones from scan to scan, which in turn, results in a non-linear variation of the removal function for this method. To solve this problem, this paper experimentally compares the surface profile formed by the laser repetitive scan zone after HF acid etching at scan time intervals of 0.1 and 30 s, respectively. It is found that the range of the fictive temperature regulated zone increases continuously with the number of laser scans when the time interval is quite short. This phenomenon can lead to the formation of non-uniform surface profiles with the same laser scanning parameters when fabricating continuous profiles. To understand the mechanism behind this phenomenon, numerical simulations were carried out to investigate the effect of scan time interval on the uniformity of the fictive temperature regulated zone under multiple laser scans. It was found that only when the laser was repeatedly scanned at time intervals of more than 1 s did the material have sufficient time to "fully absorb" the residual heat, in order that the base temperature of the material did not cause an expansion of the fictive temperature regulation zone during the next laser irradiation. The results of the numerical simulations show a good agreement with the experimental ones, which is valuable for understanding the law on the influence of residual heat on the fictive temperature under repeated laser scanning. Furthermore, it is instructive for the application of this new method to CPP processing.

**Author Contributions:** Conceptualization, W.L., X.Y., W.Z. and Q.Z.; methodology, W.L., C.Z., Y.B. and X.J. (Xiaodong Jiang); software, W.L.; validation, W.L.; investigation, J.C., K.Y., L.Z. and X.J. (Xiaolong Jiang); data curation, H.W.; visualization, X.L. All authors have read and agreed to the published version of the manuscript.

**Funding:** This research was funded by National Natural Science Foundation of China [Grant No. 62275235].

**Data Availability Statement:** The data presented in this study are available on request from the corresponding author. The data are not publicly available due to the fact that certain data relate to the subsequent work to be published.

**Acknowledgments:** The authors deeply appreciate Sun Lan from University of Science and Technology of China for providing language assistance.

**Conflicts of Interest:** The authors declare no conflict of interest.

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
