# Peer review of "Evolution Regularity of Continuous Surface Structures Shaped by Laser-Supported Fictive-Temperature Modifying"

_crystals, doi:10.3390/cryst13030542_

Round 1

Reviewer 1 Report

This work lacks the breadth and depth of research expected for publication in a journal like Crystals. The experimental tests carried out lack novelty or complexity at a scientific level, nor do they present a minimum extension. As a consequence, the results lack importance or interest for the potential reader. The suggestion of the reviewer is to reject the draft.

Reviewer 2 Report

Although the trials were conducted under very limited conditions, the combination of Co2 laser machining or heating and HF etching is a new attempt to produce groove texture patterns for manufacturing CPP (continuous phase plates). However, the following points should be addressed:

(1) Figure 2 evaluates the cross-sectional shape and dimensions of one groove per scan. But, it is necessary to show how much the groove shape and dimensions vary with longitudinal position. This is important because it concerns the quality of the CPP.

(2) The controllability of groove shape and dimensions should also be addressed. Also, an explanation of the prospects for actually producing the CPP needs to be added.

(3) Please add an explanation of what the crystal structure is like in the areas that cool down after exceeding 1300 K and eventually become more easily etched by HF.

Reviewer 3 Report

1.     Figure 3. It is not clear what 10% and 30% indicate in the legends. A short notes should be written in the captions.

2.     Figure 4. Simulation parameters, boundary conditions, equations, and software should be described in the paper.

3.     Figure 5(g). What is the cross-section for the temperature profiles demonstrated?

4.     Figure 6. It is not clear how the normalized temperature integral was calculated to obtain Figure 6(c). Some descriptions should be added in the paper.

Round 2

Reviewer 1 Report

As already indicated in the first review, this work lacks the breadth and depth of research expected for publication in a journal such as Crystals. The experimental tests carried out lack novelty or complexity at a scientific level, nor do they present a minimum extension. As a consequence, the results lack importance or interest for the potential reader. The reviewer's suggestion is to reject the revised draft (the changes made have been minimal and, in any case, no changes have been made with an impact on the scope of the research carried out).

Author Response

Thank you and the other members of the Editorial Board for your helpful suggestions. As suggested, additions have been made in the revised manuscript to further clarify the novelty of this manuscript so that the reader can more clearly understand the innovative aspects of this paper. The changes in the revised manuscript have been marked with a green background.

In any case, I would like to express my sincere thanks to the three reviewers for their careful work on this manuscript. I would also like to thank Gillbert Li, the editor responsible for this manuscript, for his professional assistance.

Reviewer 3 Report

The authors have adequately responded to the reviewers' comments.

Author Response

(The authors gave the same response as above.)
